# Molecularly Targeted Therapy in Acute Myeloid Leukemia: Current Treatment Landscape and Mechanisms of Response and Resistance

**DOI:** 10.3390/cancers15051617

**Published:** 2023-03-06

**Authors:** Curtis A. Lachowiez, Courtney D. DiNardo, Sanam Loghavi

**Affiliations:** 1Department of Medicine, Division of Hematology/Oncology, Knight Cancer Institute, Oregon Health & Science University, Portland, OR 97239, USA; 2Department of Leukemia and Hematopathology, The University of Texas, MD Anderson, Houston, TX 77030, USA

**Keywords:** acute myeloid leukemia, IDH inhibitor, FLT3 inhibitor, targeted therapy

## Abstract

**Simple Summary:**

Molecularly targeted therapy is now a standard treatment option for patients with acute myeloid leukemia and *IDH1/2* or *FLT3* mutations. This review summarizes the current treatment landscape of IDH and FLT3 inhibitors and emerging targeted therapies for the treatment of AML.

**Abstract:**

Treatment for acute myeloid leukemia (AML) has evolved rapidly over the last decade as improved understanding of cytogenetic and molecular drivers of leukemogenesis refined survival prognostication and enabled development of targeted therapeutics. Molecularly targeted therapies are now approved for the treatment of *FLT3* and *IDH1/2*-mutated AML and additional molecularly and cellularly targeted therapeutics are in development for defined patient subgroups. Alongside these welcome therapeutic advancements, increased understanding of leukemic biology and treatment resistance has resulted in clinical trials investigating combinations of cytotoxic, cellular, and molecularly targeted therapeutics resulting in improved response and survival outcomes in patients with AML. Herein, we comprehensively review the current landscape of IDH and FLT3 inhibitors in clinical practice for the treatment of AML, highlight known resistance mechanisms, and discuss new cellular or molecularly targeted therapies currently under investigation in ongoing early phase clinical trials.

## 1. Introduction

Acute myeloid leukemia (AML) is an aggressive hematologic malignancy characterized by arrested differentiation of hematopoietic stem and progenitor cells resulting from alterations in a heterogeneous array of genomic drivers [1]. Characterization of this genomic diversity led to the recognition and refinement of prognostic and predictive molecular markers, enabling risk stratification in AML [1,2]. The most recent iteration of the European LeukemiaNet (ELN) guidelines recognizes not only the importance of genomic characterization in assignment of disease risk, but the additional role of molecularly targeted therapy (namely IDH and FLT3 inhibitors) in the treatment of patients with AML [2].

AML treatment historically relied on anthracycline- and cytarabine-based regimens [3,4], however since 2017 the treatment landscape has expanded to include eleven additional FDA-approved medications and/or regimens, including eight that target specific molecular or cellular subgroups, as shown in Figure 1. Though not specific for a molecular or cellular target, CPX-351 demonstrated efficacy in patients with secondary AML or AML with myelodysplasia-related changes and oral azaciditine (CC-486) improved survival in patients with *NPM1* and/or *FLT3*-mutated AML, further supporting the ability to individualize therapy based on clinical characteristics in addition to cytogenetic or molecular features [5,6,7].

While consolidative hematopoietic cell transplantation in remission remains the only curative therapy for AML, these new treatment options are a welcome addition to a therapeutic repertoire that has largely relied upon standard cytotoxic chemotherapy for the last 40 years [3,8,9,10,11,12]. Despite these recent advances, cures remain elusive, and relapse is the most common cause of mortality [3,8,9,10,11,12]. An increased understanding of the cellular mechanisms portending sensitivity and resistance to specific therapies has prompted clinical investigations using combinations of targeted therapies in attempts to improve outcomes.

Herein, we review the contemporary landscape of molecularly targeted therapies in AML with a focus on FLT3 and IDH inhibitors, identified mechanisms of resistance to these agents, and early investigations of therapies utilizing molecularly targeted agents in combination or with the addition of cytotoxic chemotherapy.

## 2. FLT3 Mutations in AML

Mutations within the juxtamembrane domain (most commonly internal tandem duplications [ITD] and/or the tyrosine kinase domain [TKD]) of fms-related receptor tyrosine kinase 3 (*FLT3)* are identified in approximately 30% (ITD: ~20%, TKD: ~7%) of patients with newly diagnosed AML [13,14,15].

*FLT3*- ITD mutations are associated with leukocytosis, lower response rates, and increased relapse risk correlating with shorter event-free (EFS) and overall survival (OS) in patients receiving intensive chemotherapy (IC) [16]. *FLT3*-TKD mutations impart less influence on prognosis in patients treated with IC or lower-intensity therapies containing venetoclax (VEN) [17,18]. Given the adverse prognosis associated with *FLT3*-ITD mutations in patients treated with IC, European LeukemiaNet (ELN) 2017 guidelines classified AML as favorable-, intermediate-, or adverse-risk dependent upon co-mutations in *NPM1*, and the *FLT3*-ITD allelic ratio (AR: mutated alleles/wild-type alleles) [19]. The recently updated ELN 2022 guidelines omit the allelic ratio, in part due to improvements in FLT3 directed therapy, and now classify all patients with *FLT3*-ITD mutations as intermediate-risk [20].

Multiple FLT3 inhibitors (FLT3i) have been developed to target FLT3, which broadly can be classified as type 1 (inhibit both ITD and TKD variants) or type 2 (inhibit only ITD variants). Prospective studies of FLT3i monotherapy or in combination with chemotherapy are displayed in Table 1.

### 2.1. Sorafenib Combined with IC

Sorafenib, a type 2 multi-kinase FLT3i was evaluated in the randomized, double-blind, placebo-controlled phase 2 SORAML trial [21]. Patients aged ≤60 with newly diagnosed AML received standard IC (daunorubicin plus cytarabine; the so-called “7+3” regimen) followed by cytarabine consolidation with or without sorafenib (400 mg BID) administered on days (D) 10–19 during induction and D8–28 in consolidation, followed by up to 12 months of maintenance therapy [21]. Notably, the study population was not limited to patients with *FLT3* mutations (only 17% of patients in each arm had a *FLT3* mutation) [21].

Complete response (CR) rates were similar between groups (60% vs. 59%). Sorafenib improved EFS (3-year EFS: 40% vs. 22%, *p*-value: 0.013), and reduced the incidence of relapse (3-year cumulative incidence of relapse: 34% vs. 49%, *p*-value: 0.033), albeit with no significant difference in OS (adjusted hazard ratio [HR]: 0.84 [95% confidence interval [CI]: 0.59–1.19], *p*-value: 0.322) [21]. Long-term follow-up of approximately 6.5 years confirmed the EFS benefit of sorafenib (HR: 0.61 [95% CI: 0.44–0.87], *p*-value: 0.006) without an OS benefit (HR: 0.74, [95% CI: 0.49–1.12]) [22]. Increased toxicity was observed with the addition of sorafenib; grade 3 or greater adverse events (AE) that were more common in the sorafenib arm included fever, diarrhea, bleeding, cardiac events, hand-foot syndrome, and rash [21].

### 2.2. Sorafenib Maintenance Post-Allogeneic Stem Cell Transplantation

Sorafenib has also been investigated as post-transplant maintenance therapy in patients with *FLT3*-ITD-mutated AML undergoing allogeneic hematopoietic cell transplantation (HCT). In the randomized, placebo-controlled, double-blind phase 2 SORMAIN trial, 83 patients with *FLT3*-ITD-mutated AML in CR (71% in CR_1_, 28% outside CR_1_) following HCT were assigned to receive either sorafenib (maximum dose of 400 mg twice daily [BID]) or placebo for two years [23]. After a median follow-up of approximately 42 months, sorafenib significantly reduced the risk of relapse or death (HR: 0.39 [95% CI: 0.18–0.85], *p*-value: 0.013) compared to placebo, corresponding to a 24-month relapse free survival of 85% vs. 53.3%. The 24-month OS was 90.5% vs. 66.2% (*p*-value: 0.007) and a trend towards improved median OS was observed in the sorafenib arm (*p*-value: 0.085) [23]. Graft-vs-host-disease (GVHD) was common in both arms (sorafenib arm: 76.8% vs. placebo: 59.8%). Post-HCT dose reductions occurred in 48.8% and 40% of patients receiving sorafenib and placebo, respectively [23].

### 2.3. Midostaurin Combined with IC

The multicenter randomized, double-blind, placebo-controlled phase 3 RATIFY trial evaluated the orally available type 1 FLT3i midostaurin combined with IC in younger (age < 60) patients with newly diagnosed (ND) AML and *FLT3*-ITD and/or TKD mutations [12]. Treatment consisted of standard anthracycline- and cytarabine-based induction with midostaurin 50 mg orally [PO] BID, administered on D8–21 every 28 days. Consolidation employed high-dose cytarabine (3000 mg/m^2^) on D1, D3, and D5 with administration of midostaurin on D8–21 every 28 days up to four cycles. Midostaurin could then be administered continuously as maintenance therapy for up to 12 cycles (approximately 1 year) [12].

Similar CR rates were observed with midostaurin vs. placebo (CR: 58.9% vs. 53.5%, *p*-value: 0.15). Midostaurin improved OS compared to placebo (HR: 0.78 [95% CI: 0.63–0.96], one-sided *p*-value: 0.009), corresponding to a ~7% increase in OS (4-year OS: 51.4% vs. 44.3%) [12]. Midostaurin also reduced the risk of remission failure, relapse, or death compared to placebo (HR: 0.78 [95% CI: 0.66–0.93]; one-sided *p*-value: 0.002). The benefit of midostaurin was observed across all *FLT3*-mutated patients (i.e., including patients with *FLT3*-ITD and/or *TKD* mutations; a trend in favor of midostaurin was observed in subgroup analyses of individual *FLT3* variants) consistent with its known activity against either FLT3 variant.

Fifty-seven percent of patients underwent HCT, with a trend towards improved survival with midostaurin when performed in CR_1_ (median OS was not reached in either group, *p*-value: 0.07). A sensitivity analysis censored for OS at the time of HCT demonstrated 4-year OS rates of 63.7% vs. 55.7% with midostaurin vs. placebo, respectively (*p*-value: 0.08). No significant differences in grade 3 or greater AEs were observed between arms, other than an increased risk of rash in patients treated with midostaurin compared to placebo (14% vs. 8%, *p*-value: 0.008) [12].

## 3. Gilteritinib

The second-generation type 1 FLT3i gilteritinib demonstrated efficacy as monotherapy in patients with relapsed/refractory (R/R) AML compared to salvage chemotherapy in the randomized phase 3 ADMIRAL trial [8]. A total of 371 patients enrolled and were randomized to gilteritinib 120 mg PO daily, administered continuously or the investigators’ choice of salvage chemotherapy.

Thirty-four percent of patients treated with gilteritinib attained a CR or CR with partial hematologic recovery (CRh) compared to 15.3% of patients assigned to the control arm. Patients with *FLT3*-ITD-mutated AML treated with gilteritinib demonstrated a CR rate of 20.5% vs. 9.7% when treated with salvage chemotherapy.

After a median follow-up of 17.8 months, gilteritinib improved median OS compared to the control arm (9.3 vs. 5.6 months, *p*-value < 0.001); while median EFS was not significantly different (2.8 vs. 0.7 months; HR: 0.79 [95% CI: 0.58–1.09]). Gilteritinib improved OS compared with salvage chemotherapy in the subgroup of patients with co-mutations in *NPM1* and *DNMT3A* (median OS 10.8 vs. 5.0 months), a molecular subgroup with historically poor outcomes [8,24,25].

Common grade 3 or greater adverse events occurring with gilteritinib were febrile neutropenia (45.9%), anemia (40.7%), and thrombocytopenia (22.8%) [8]. Serious adverse events attributed to gilteritinib included febrile neutropenia (9.3%) and elevated liver function tests (LFTs; alanine aminotransferase [ALT] elevated: 4.5%, aspartate aminotransferase [AST] elevated: 4.1%). Eleven percent of patients discontinued gilteritinib therapy in relation to adverse events. The 30 and 60-day mortality with gilteritinib was 2% and 10.2% vs. 7.7% and 19% with salvage chemotherapy, respectively. Thus, in the relapsed setting, gilteritinib monotherapy was more effective with lower early mortality compared to salvage chemotherapy.

## 4. Quizartinib

Quizartinib was evaluated in a multicenter, randomized phase 3 study investigating the type 2 FLT3i quizartinib (60 mg daily) versus salvage chemotherapy in patients with R/R *FLT3*-ITD-mutated AML with a prior remission duration of 6 months or less [26]. Quizartinib resulted in a composite CR rate of 48% vs. 27% and significantly improved OS (median OS 6.2 vs. 4.7 months, *p*-value: 0.02) compared to salvage chemotherapy [26]. No significant difference was observed with respect to EFS between quizartinib vs. salvage chemotherapy (median 1.4 vs. 0.9 months, *p*-value: 0.11). Adverse events were similar between the two study arms. Grade 3 or greater cardiac events with quizartinib were infrequent, with grade 3 QT prolongation occurring in 4% of patients, 5% of patients had QT prolongation necessitating quizartinib interruption, and two patients discontinued quizartinib secondary to QT prolongation [25]. Currently, quizartinib is approved only in Japan for the treatment of R/R FLT3-ITD-mutated AML.

### Quizartinib Combined with IC

Quizartinib was evaluated in a recently completed, randomized phase 3 trial of IC combined with quizartinib compared with IC in patients with ND-AML and a *FLT3*-ITD mutation [27]. Treatment consisted of standard anthracycline- and cytarabine-based induction and consolidation, with the addition of quizartinib 40 mg daily on D8–21 during induction and D6–19 of consolidation [27]. The addition of quizartinib modestly improved CR/ complete response with incomplete hematologic recovery (CRi) rates (71.6% vs. 64.9%), however resulted in a significant OS benefit compared to placebo (median OS: 31.9 vs. 15.1 months, *p*-value: 0.03) [27], with approximately a third of patients in both arms (quizartinib: 31%, placebo: 27%) proceeding with HCT. More grade 3 or greater neutropenia occurred in the quizartinib arm (18.1% vs. 8.6%), and more patients in the quizartinib arm discontinued treatment due to adverse events (20.4% vs. 8.6%) [27]. Treatment-emergent adverse events resulting in death occurred in 11.3% of the quizartinib arm compared to 9.7% of the placebo arm, and grade 3 or greater QT prolongation was infrequent with quizartinib (3.0%) [27].

## 5. Ongoing Investigations of FLT3-Inhibitors with Intensive Chemotherapy

Additional investigations assessing if outcomes in *FLT3*-mutated AML in combination with IC can be further improved through augmentation of the chemotherapy backbone, use of an alternative FLT3i, or both are ongoing.

A recent update of the IC regimen consisting of cladribine, cytarabine, and idarubicin (CLIA) combined with gilteritinib in 24 patients reported a CR/CRi rate of 75%, with a median OS that had not been reached at the time of data presentation [28]. A randomized phase 2 study evaluating the efficacy of gilteritinib vs. midostaurin combined with 7+3 is ongoing (NCT03836209). Long-term follow-up of a phase 2 study reporting the efficacy of the type 1 FLT3i crenolanib in combination with 7+3 reported a CR/CRi rate of 86% with 50% (N = 22) of patients transitioning to HCT [29]. After a median follow-up of 45 months, median EFS was 45 months, and median OS had not been reached (OS at time of data analysis was 57%) [29]. Based upon these results, an ongoing phase 3 trial comparing 7+3+crenolanib vs. 7+3+midostaurin is ongoing (NCT03258931). As multiple FLT3i are active in the frontline setting with unique side effect profiles, treatment selection will be dependent not only upon efficacy, but also tolerability and (in the case of quizartinib) co-occurring *FLT3*-TKD mutations.

### 5.1. Lower-Intensity Chemotherapy with FLT3-Inhibitors

While a benefit of FLT3i therapy has been observed with intensive chemotherapy, the role of FLT3i combined with lower-intensity therapies including the hypomethylating agent azacitidine (AZA) remains unclear [30].

The LACEWING trial was a randomized, phase 3, open-label investigation evaluating the efficacy of frontline therapy with gilteritinib (Gilt; administered 120 mg PO daily) in combination with AZA vs. a placebo combined with AZA [30]. The population included older patients (median age Gilt+AZA vs. AZA: 78 vs. 76 years) unfit to receive IC [30]. Most patients had isolated *FLT3*-ITD mutations (Gilt+AZA vs. AZA: 78.4% vs. 81.6%) with intermediate-risk cytogenetics (Gilt+AZA vs. AZA: 68.9% vs. 73.5%).

Similar CR/CRh rates were observed with Gilt+AZA vs. AZA (25.7% vs. 16.3%, *p*-value: 0.21). After a median follow-up of 9.76 months, no significant difference in the primary outcome of OS was observed (median OS: 9.82 vs. 8.87 months, *p*-value: 0.75), resulting in an early termination of the study [30]. The median EFS also did not differ between treatment arms (HR: 0.925 [95% CI: 0.59–1.44], *p*-value: 0.839).

Serious adverse events (SAEs) were more common with Gilt+AZA vs. AZA (87.7% vs. 63.8%), most commonly febrile neutropenia and pneumonia. Of note, gastrointestinal hemorrhage occurred in 12.3% and 6.4% of patients treated with Gilt+AZA vs. AZA, respectively. AEs resulting in death occurred in 26% and 23.4% of patients receiving Gilt+AZA vs. AZA, respectively.

In contrast to the LACEWING study results, a single-institution, retrospective cohort analysis of 16 patients (median age 71 years) treated with lower-intensity therapies combined with quizartinib reported a median OS of 15.7 months, which compares favorably to patients with *FLT3*-ITD mutations treated with AZA+VEN [31,32]. However given the largely negative findings of the LACEWING study, future prospective investigations are necessary to define the role of FLT3i combined with lower-intensity therapy for the frontline treatment of AML.

### 5.2. Venetoclax Combined with Gilteritinib

FLT3i demonstrate synergy through alterations of alternative anti-apoptotic expression when used in combination with VEN, a potent inhibitor of the anti-apoptotic B-cell lymphoma-2 (BCL-2) protein [33,34,35].

The combination of gilteritinib and VEN (VenGilt) demonstrated efficacy in *FLT3*-ITD-mutated R/R-AML in a phase 1b multicenter, open label study (NCT03625505) of predominantly older (median age 63 years) patients [33]. Sixty-four percent of patients received a prior FLT3i, and sixteen percent, a prior VEN. VenGilt administered orally daily for 28-day cycles resulted in a modified composite CR (mCRc) rate (CR+CRi+complete response with incomplete platelet recovery [CRp]+morphologic leukemia free state [MLFS]) of 75% (N = 42/56); the corresponding CRc rate (CR/CRi/CRp) was 39% [33]. The median OS was 10.6 months and 9.6 months in the subgroup of patients without and with prior FLT3i exposure; median OS was 6.7 months in patients with prior VEN exposure [33]. Sixty percent of evaluable patients (N = 25) attained a molecular response (i.e., *FLT3*-ITD variant allele frequency [VAF] < 10^−2^), corresponding to a median OS of 11.6 months [33].

Adverse events with VenGilt were notable. Ninety-seven percent of patients experienced a grade 3/4 AE, and 46% experienced an SAE [33]. The most common grade 3/4 AEs were related to myelosuppression, with 80% of patients experiencing grade 3/4 cytopenias [33]. Dose interruptions for VEN and Gilt were required in 13% and 8% of patients, respectively [33].

Given the high rate of patients receiving prior FLT3i (64%), these results compare favorably to the ADMIRAL trial, where 13% of patients assigned to the gilteritinib arm (median OS 9.3 months) received a prior FLT3i [8,33]. As the most prominent AEs related to myelosuppression, caution must be exerted when using this regimen with active monitoring of peripheral blood counts and judicious dose adjustment on behalf of the provider.

### 5.3. Triplet Regimens in FLT3-Mutated AML

Single-arm, single-institution phase 1/2 trials evaluating triplet combinations of FLT3i combined with lower-intensity therapies and VEN have reported promising outcomes [32,36]. A retrospective analysis of older adults (median age 69 years) with *FLT3*-mutated AML treated with a FLT3i and a hypomethylating agent (azacitidine or decitabine) combined with VEN reported increased CR/CRi rates (93% vs. 70%), measurable residual disease (MRD) negativity rates (*FLT3*-ITD polymerase chain reaction: 96% vs. 54%, *p*-value < 0.01; multiparameter flow cytometry [MFC]: 83% vs. 38%, *p*-value < 0.01), and OS (median not reached vs. 9.5 months, *p*-value < 0.01) compared to patients treated with doublet regimens combining a FLT3i with lower-intensity chemotherapy [32]. These results also compare favorably to reported CR/CRi rates (*FLT3*-ITD: 63%, *FLT3-*TKD: 76.9%) and median OS (*FLT3*-ITD: 9.9 months; *FLT3*-TKD: 19.2 months) in patients with *FLT3*-mutated AML treated with AZA+VEN [31].

Triplet regimens were more myelosuppressive vs. doublet regimens following cycle 1, with a median time to absolute neutrophil count [ANC] recovery ≥ 500 μL of 40 vs. 20 days, albeit this result was not statistically significant (*p*-value: 0.15) [32]. Prospective investigations of gilteritinib and quizartinib in combination with azacitidine or decitabine and venetoclax are ongoing to better inform outcomes in this setting (NCT04140487; NCT03661307) [37,38].

### 5.4. FLT3 Mechanisms of Resistance

Both primary (i.e., refractory disease) and secondary (i.e., relapse after response) resistance have been described in FLT3i therapy, as shown Figure 2 [39].

Next generation sequencing (NGS) of paired diagnosis and relapse samples in 173 patients treated with type 1 or type 2 FLT3i identified mutations within the RAS/MAPK pathway in a significant portion of patient samples demonstrating primary or secondary resistance [39].

Patients with refractory disease more frequently had a higher clonal burden of *RAS* mutations (median VAF: 31% vs. 6%) at diagnosis, albeit these results did not reach statistical significance (*p*-value: 0.19) [39]. In patients treated with type 1 FLT3i with activity against both *FLT3*-ITD and TKD mutations, a *RAS* VAF of ≥20% was more often identified in non-responding vs. responding patients (*p*-value: 0.023) [39]. Of importance, 58% (N = 39) of the primary resistance cohort had R/R-AML; whether these results can be extrapolated to ND-AML patients is unclear.

In patients with secondary resistance, 55% (N = 33/67) of patients experiencing disease relapse following prior FLT3i therapy had a newly identifiable mutation, which was more commonly observed in patients treated with type 2 FLT3i therapy compared to type 1 FLT3i (65% vs. 33%, *p*-value: 0.02) [39]. Common mutations identified at relapse included *FLT3*-D835 (21%), RAS/MAPK pathway mutations (*NRAS*, *PTPN11*, *CBL*; 13%), *IDH1* or *IDH2* (9%), *WT1* (7%), and *TP53* (7%) [39]. Mutations in the *FLT3*-TKD domain (i.e., *FLT3*-D835) were common in patients receiving type 2 FLT3i therapy (30%), while *RAS* pathway genes (29%) were common in patients treated with type 1 FLT3i therapy [39].

Oncogenic signaling appears to correlate with FLT3i resistance irrespective of which type 1 FLT3i is utilized [40]. In an analysis of 59 patients with R/R-AML treated with gilteritinib, mutations in RAS/MAPK pathway genes were identified in approximately-one third (37%) of patients, including *NRAS*, *KRAS*, *PTPN11*, *CBL*, and *BRAF* [41]. Similarly, in patients treated with midostaurin without a detectable *FLT3*-ITD mutation at relapse, RAS/MAPK signaling mutations were also observed.

Single-cell DNA sequencing (scDNAseq) demonstrated gilteritinib therapy led to the selection of resistant *NRAS*-mutated clones with or without co-occurring *FLT3* mutations that expanded at relapse [41]. In vitro and murine models demonstrated combining a MEK inhibitor with gilteritinib partially abrogated *NRAS* mediated relapse, providing evidence that combined therapy targeting the RAS/MAPK pathway may enhance leukemic cell death [41,42].

Additional mechanisms of resistance to gilteritinib include the outgrowth of leukemic clones containing alternative mutations in genes outside the RAS/MAPK pathway including *SF3B1, IDH1*, and the development of *BCR::ABL1* fusions at the time of relapse [41,43,44]. Similar molecular patterns have been observed in patients treated with midostaurin [40]. Sequencing of *FLT3*-mutated R/R-AML samples revealed the acquisition of mutations in alternative genes aside from RAS/MAPK pathway regulators including *IDH1*, *WT1*, *RUNX1, ASXL1*, *SF3B1*, *U2AF1,* and *ZBTB7A* [40].

Varying cellular resistance mechanisms and resistance intrinsic to *FLT3* have also been described within the context of specific FLT3i treatment [40,41,42]. Mutations including *FLT3* F691L (a known gatekeeper mutation that impairs gilteritinib binding) and *FLT3* N701K confer resistance to gilteritinib and may be overcome by higher gilteritinib doses, while mutations in in *FLT3* N676 confer resistance to midostaurin [40,41,42]. Changes within *FLT3* clones and ITD insertion sites between diagnosis and relapse have been identified in patients following midostaurin therapy [40]. Upregulation of alternative kinases including aurora kinase B (AURKB), effect of tumor microenvironment on *FLT3* clone sensitivity, and metabolic reprogramming have all been implicated as additional mechanisms resulting in early or late resistance to gilteritinib [42].

## 6. Isocitrate Dehydrogenase Mutations in AML

Mutations within the isocitrate dehydrogenase 1 or 2 genes (i.e., *IDH1* or *IDH2*) occur in approximately 20% (*IDH1*: ~7–8%; *IDH2*: ~9–12%) of ND-AML [45,46,47]. Amino acid changes at the R132 residue in *IDH1* or R140/R172 residue in *IDH2* result in formation of neomorphic enzymatic activity and production of the oncometabolite 2-hydroxyglutarate (2-HG) [48,49]. Accumulation of 2-HG impairs alpha-ketoglutarate-dependent enzymes (including *TET2*) altering epigenetic regulation and cellular metabolism resulting in impaired cellular differentiation and proliferation- thereby promoting leukemogenesis [50,51,52]. *IDH-*mutated leukemia cells have an altered mitochondrial outer membrane permeability threshold, increasing their reliance on the antiapoptotic protein BCL2 for cell survival [53].

Conflicting results have been reported on the prognostic impact of *IDH* mutations in AML. Some analyses report a negative prognostic impact while others report a neutral, or favorable prognostic influence of *IDH* mutations dependent upon co-occurring cytogenetic changes and gene mutations [46,54,55,56,57]. As with FLT3i, the development of IDH inhibitors (IDHi) resulted in targeted therapy for a molecularly defined subgroup of AML. Currently, three IDHi are approved for the treatment of *IDH1* or *IDH2*-mutated AML. Prospective studies of IDH inhibitors are displayed in Table 2.

### 6.1. Ivosidenib in IDH1-Mutated AML

Ivosidenib (IVO), an orally available IDH1 inhibitor is approved for the treatment of patients with newly diagnosed and relapsed/refractory AML [9]. In an open label, phase 1 study that included a population of older patients (median age 76.5 years) with ND-AML unfit for IC, ivosidenib resulted in a CR/CRh rate of 42%, and a median OS of 12.6 months [9]. IVO effectively targeted *IDH1*-mutated leukemic clones, resulting in *IDH1* clearance in 64% of patients attaining a CR/CRh [58]. After a median follow-up of 24 months, the median duration of response was not reached [9].

In patients with R/R-AML IVO resulted in CR/CRh rate of 30% (including a true CR rate of 22%) [9]. After a median follow-up of approximately 15 months, the median OS was 8.8 months. Mutations in signaling pathway genes were frequent in non-responding patients. In the R/R-AML population, *IDH1* mutation clearance occurred in 21% of patients attaining CR/CRh, compared to no patients with a response less than CR/CRh. *IDH1* mutation clearance correlated with a median duration of remission and OS of 11.1 and 14.5 months, respectively [9].

Grade 3 or higher adverse events at the recommended dose of 500 mg daily were uncommon with the most frequent being QTc prolongation (7.8%), IDH-differentiation syndrome (IDH-DS; 3.9%), and anemia (2.2%). Other common adverse events occurring in 20% or more of the study population included diarrhea, leukocytosis, febrile neutropenia, fatigue, dyspnea, peripheral edema, anemia, fever, and cough [9].

### 6.2. Ivosidenib with Azacitidine in IDH1-Mutated AML

The multicenter prospective, randomized phase 3 AGILE trial compared IVO combined with AZA (IVO+AZA) vs. AZA in 146 patients with newly diagnosed, *IDH1*-mutated AML [59]. The combination of IVO+AZA improved CR/Cri rates (51% vs. 18%, *p*-value < 0.001), EFS (HR: 0.33 [95% CI: 0.16–0.69], *p*-value: 0.002), and OS (24 vs. 7.9 months, *p*-value: 0.001) compared to AZA. *IDH1* mutation clearance occurred in 52% vs. 30% of patients receiving IVO+AZA vs. AZA, respectively [59]. As with observations with single-agent IVO, baseline *IDH1* VAF did not correlate with response [9,58,59].

IVO+AZA resulted in a notably lower rate of infectious AEs compared to AZA (28% vs. 49%), but a higher rate of hemorrhagic events (41% vs. 29%). Adverse events of special interest, including IDH-DS, occurred in 14% of patients in the IVO+AZA arm. While neutropenia more commonly led to drug interruption with IVO+AZA vs. AZA (23% vs. 4%), rates of febrile neutropenia (10% vs. 8%) and pneumonia (8% vs. 7%) were similar.

### 6.3. Olutasidenib in IDH1-Mutated AML

Olutasidenib (OLUTA; FT-2102), an allosteric non-competitive inhibitor of mutant IDH1 also demonstrated safety and efficacy with (N = 46) or without AZA (N = 32) in a phase 1/2 study of patients with *IDH1*-mutated AML or MDS [5,60]. OLUTA monotherapy resulted in a CR/CRh in 32% (N = 7/22) of patients with R/R-AML, while OLUTA+AZA resulted in a CR/CRh rate of 15% (N = 4/26) of patients with R/R-AML, and 54% (N = 7/13) in patients with ND-AML, similar to the CR/CRi rate observed with IVO+AZA in ND-AML (CR/CRi: 51%) [60]. The median time to response was 1.9 months irrespective of regimen and the median duration of response in ND-AML patients treated with OLUTA+AZA was not reached [60].

The median follow-up was 6.7 and 9.3 months for patients treated with OLUTA or OLUTA+AZA, respectively. In patients with R/R-AML, median OS was 8.7 months with OLUTA, and 12.1 months with OLUTA+AZA; ND-AML patients experienced a median OS of 8.8 months and not reached (estimated 12-month OS: 75%) when treated with OLUTA or OLUTA+AZA, respectively [60]. In responding patients with AML, treatment with OLUTA with or without AZA resulted in a reduction in *IDH1* VAF to < 1% in 40% of patients [60].

Grade 3 or 4 adverse events of special interest in the phase 1 portion of the study included IDH-DS in 13% of patients receiving OLUTA or OLUTA+AZA, LFT abnormalities in 16% and 11% of patients receiving OLUTA and OLUTA+AZA, and QT prolongation in 7% of patients receiving OLUTA+AZA. Rates of ≥grade 3 infectious complications were similar (37% vs. 33%) between patients treated with OLUTA or OLUTA+AZA [60]. Febrile neutropenia occurred in 22% vs. 32% of patients treated with OLUTA vs. OLUTA+AZA; AEs leading to treatment discontinuation occurred in 28% of the monotherapy group vs. 17% of the combination group [60].

Updated data from the ongoing, multicenter, open label phase 2 cohort study of OLUTA (150 mg BID) in patients with R/R-AML was recently reported [5]. In the primary efficacy population comprised of 147 patients with R/R-AML, OLUTA resulted in a CR/CRh rate of 35%, confirming the results observed in earlier cohorts [5]. Of interest, 33% of patients who received prior VEN-based therapy (N = 4/12) attained CR/CRh. The median duration of response in patients attaining CR/CRh was 25.9 months, which compares favorably to the median duration of CR/CRh observed with IVO (8.2 months) [9]. The median OS was 11.6 months in the overall study population. An 18-month survival time was estimated to be 78% in patients attaining CR/CRh, with *IDH1* VAF clearance observed in 28% (N = 11/39) [5].

No new safety signals emerged within this expanded R/R-AML cohort. Whether the improved durability of response with OLUTA compared to IVO in R/R-AML is due to its allosteric inhibition of mutant IDH1 or through suppression of resistance mechanisms observed with IVO remains unknown.

### 6.4. Enasidenib in IDH2-Mutated AML

Enasidenib (ENA), an orally available IDH2 inhibitor, is approved for use in patients with R/R-AML and a mutation in *IDH2* [10]. In the efficacy evaluable population (N = 176) of an open label, phase 1/2 trial, ENA resulted in an overall response rate (ORR; CR+CRi+CRh+partial remission [PR]+ MLFS) of 40.3% with a CR rate of 19.3% [10]. Slight improvement in ORR (35.4% vs. 53.3%) and CR (17.7% vs. 24.4%) rates were observed between patients with *IDH2* R140 vs. *IDH2* R172 mutations [10]. After a median study follow-up of 7.7 months, median EFS and OS were 6.4 and 9.3 months, respectively. The median OS was 19.7 months in patients attaining a CR [10].

In the safety population treated at the FDA-approved dose of enasidenib (100 mg daily; N = 153) common treatment-related AEs included hyperbilirubinemia (8%) due to known hepatic inhibition of UGT1A1, IDH-DS (11%), anemia (10%), thrombocytopenia (8%), and TLS (5%) [10]. Treatment with ENA resulted in myeloid differentiation with evidence of mature myeloid cells harboring *IDH2* mutations in addition to other clonal cytogenetic changes, consistent with its mechanism of action as a differentiation agent [10].

In a confirmatory open label, randomized phase 3 trial comparing ENA (N = 158) to conventional care regimens (CCR; azacitidine [N = 69], low-dose cytarabine [N = 37], intermediate-dose cytarabine [N = 33], or best supportive care only [N = 22]) in patients with *IDH2*-mutated R/R-AML, no significant difference in OS was observed with ENA vs. CCR (median OS 6.5 vs. 6.2 months, HR: 0.86 [95% CI: 0.7–1.10], *p*-value: 0.23), although a 12% difference in 12-month OS was observed in favor of the ENA arm [61]. Within a modified intention-to-treat population (defined as patients with an AML diagnosis and no eligibility criteria violations who received at least one dose of study drug and one efficacy assessment), the median OS was significantly improved with ENA vs. CCR (6.9 vs. 5.4 months, HR: 0.70 [95% CI: 0.50–0.98], *p*-value: 0.03).

In the overall study population, ENA treatment improved ORR (40.5% vs. 9.9%, *p*-value < 0.001), CRc rate (29.7% vs. 6.2%, *p*-value < 0.001), EFS (median 4.9 vs. 2.6 months, HR: 0.68 [95% CI: 0.52–0.91], *p*-value: 0.008), and percentage of patients with improvement in erythroid, neutrophil, and/or platelet counts (42.2% vs. 11.2%, *p*-value < 0.001) [61]. Thus, while the trial was negative for the primary endpoint of OS, the improvement in meaningful secondary endpoints supports the efficacy of ENA within this high-risk patient population.

ENA was also evaluated in a subset of patients with ND-AML. In a population of 39 patients, ENA resulted in an ORR of 30.8%, including a CR/CRi rate of 21% [62]. After a median study follow-up of 8.4 months median EFS and OS were 5.7 and 11.3 months, respectively [62]. Adverse events were similar to those observed in the R/R-AML population. Common adverse events ≥ grade 3 occurring in ≥10% of the patient population included hyperbilirubinemia (13%), anemia (13%), IDH-DS (10%), thrombocytopenia (8%), and tumor lysis syndrome (8%). One death occurred that was potentially related to IDH-DS [62]. No significant difference in 2-HG suppression was observed between responding vs. non-responding patients, though patients with mutations in *DNMT3A* had improved CR rates (*p*-value: 0.045) [10].

### 6.5. Enasidenib with Azacitidine in IDH2-Mutated AML

A multicenter, phase 1b/2 study investigated 107 patients with newly diagnosed *IDH2*-mutated AML treated with the combination of ENA+AZA vs. AZA monotherapy [63]. In the phase 2 portion of the study, treatment with ENA+AZA compared with AZA improved ORR (74% vs. 36%, OR: 4.9 [95% CI: 2–11.9], *p*-value: 0.003) and CR/CRi rates (63% vs. 30%, OR: 4.0 [95% CI: 1.6–9.6], *p*-value: 0.0019), corresponding with a median duration of response of 13.9 vs. 9.9 months at an updated interim analysis [63]. Hematologic improvement was observed in 71% vs. 58% of patients treated with ENA+AZA vs. AZA, respectively. The median EFS with ENA+AZA compared with AZA was 15.9 vs. 11.9 months (*p*-value: 0.11), and the median OS was 22 vs 22.3 months (*p*-value: 0.97) [63].

No new safety signals emerged with the combination of ENA+AZA, though rates of grade 3–4 neutropenia (37% vs. 25%) and thrombocytopenia (37% vs. 19%) were numerically higher with the combination compared to AZA monotherapy. While no improvement in OS was noted, ENA+AZA was a clinically active regimen resulting in improved response rates, durable remissions, and improvement in clinically meaningful hematologic parameters.

ENA+AZA was also evaluated in patients with R/R-AML [64]. In a phase 2 trial investigating the combination of ENA+AZA (which additionally allowed incorporation of VEN and/or a FLT3i as appropriate) in patients with *IDH2*-mutated ND or R/R-AML, the CR/CRi rate was 58% (N = 11/19) in patients with R/R-AML, with a true CR rate of 26% (N = 5) [64]. Notably, the CRc rate in patients treated with ENA+AZA+VEN was 86% (N = 6/7). Two patients treated with ENA+AZA+VEN attained MRD-negative remissions measured using MFC [64]. The median OS was 9.7 months in patients with R/R-AML, with improved survival observed in patients treated in first relapse compared to second or later relapses (1 year OS: 75% vs. 10%, HR: 0.24 [95% CI: 0.07–0.79], *p*-value: 0.04). Survival appeared to favor ENA+AZA+VEN compared to ENA+AZA (1-year OS: 67% vs. 20%, HR: 0.29 [95% CI: 0.09-0.97), *p*-value: 0.08), albeit the small sample size limits statistical power. Common adverse events in patients with R/R-AML included hyperbilirubinemia (37%), febrile neutropenia (26%), and diarrhea (21%) [64].

### 6.6. IDH Inhibitors Combined with Intensive Induction Therapy

IVO and ENA have both been evaluated in combination with IC, incorporating a standard anthracycline and cytarabine backbone [65]. In a multicenter phase 1 study, patients with ND-AML with an *IDH1* (N = 60) or *IDH2* mutation (N = 93) received the respective IDHi at FDA-approved doses in combination with daunorubicin 60 mg/m^2^ administered D1-3 and cytarabine 200 mg/m^2^ administered D1-7 during induction, followed by up to four consolidation cycles of intermediate/high-dose cytarabine or one cycle of mitoxantrone/etoposide [65]. IVO or ENA was administered continuously from the start of induction and was permitted to continue as maintenance therapy until disease progression, unacceptable toxicity, or HCT.

The composite CR rate (CR+CRi+CRp) for patients treated with 7+3 combined with IVO or ENA was 77% and 74%, respectively. CR rates for patients treated with IVO and ENA were 68% and 55%, respectively. The median OS was not reached (12-month OS: 78%) in the IVO cohort and was 25.6 months in the ENA cohort [65]. No mutations correlated with response in the IVO cohort. Conversely, mutations in *ASXL1, NRAS, U2AF1*, and *TP53* were associated with resistance and *DNMT3A* associated with sensitivity to ENA [65]. *IDH1* or *IDH2* mutation clearance rates in evaluable patients in CR occurred in 39% (N = 16/41) and 23% (N = 15/64) of patients, respectively. Of interest, in a subset of patients in whom concurrent *IDH1* mutation analysis and MRD-MFC assessment was performed, *IDH1* or *IDH2* clearance was observed in 60% and 90% of MRD-MFC negative patients, indicative of the molecular heterogeneity commonly observed in AML [65].

The incorporation of IDH inhibitors with IC was tolerable. No new safety signals emerged, with an adverse event profile similar to IC without an IDHi. One dose-limiting event of persistent grade 4 thrombocytopenia occurred in a patient receiving 7+3 combined with ENA [65].

### 6.7. Mechanisms of Resistance to IDH Inhibitors

Several mechanisms of resistance to single-agent IDHi have been described Figure 2 [66,67,68,69]. *IDH* intrinsic mechanisms of resistance include second-site mutations in *IDH1* or *IDH2* [69]. *IDH2* mutations Q316E and I319M within exon 7 have been identified in patients with progressive AML following treatment with ENA [69]. Notably, these mutations occurred in trans (i.e., on the opposite allele) to the originally identified R140Q mutations and at the region of interface with ENA [69]. A similar phenomenon was reported in a patient with *IDH1*-mutated AML treated with IVO, where a second-site mutation in *IDH1* (S280F) was identified in cis, conferring IVO resistance [69]. Other described second-site *IDH1* mutations include R119P, D279N, G131A, and G289D with indirect effects outside the IVO or NADPH binding pockets, and H315D with direct effects on the IVO or NADPH binding pockets [66].

Outgrowth of leukemic clones containing alternative mutant *IDH* isoforms (i.e., emergence or expansion of an *IDH2*-mutated leukemic clone in the setting of IDH1 inhibitor treatment) also drive resistance to monotherapy [66,67]. In patients treated with IVO, selective outgrowth of *IDH2* R140Q and *IDH2* R172W clones correlated with relapse and subsequent increases in 2-HG [67,68]. Outgrowth of *IDH1* R132C containing clones conferring resistance to ENA therapy was observed in a patient with relapsed *IDH2*-mutated AML [67]. Single-cell DNA sequencing in patients treated with IVO confirmed *IDH2*-mutated clones can emerge independent of *IDH1* mutation or as resistant subclones following IDHi therapy [66].

IDH extrinsic molecular mechanisms of relapse to IDHi therapy include mutations within active signaling and transcription factor mutations [66,68,70]. The presence of mutations in active signaling pathway genes (i.e., *N/KRAS*, *PTPN11*, *FLT3*-ITD/TKD, or *KIT*) corresponded to a lower CR/CRi (7% vs. 43%) following IVO monotherapy [66]. Similar resistance mechanisms were observed with ENA, where mutations in *NRAS*, *PTPN11*, and *FLT3* were enriched in non-responding patients [68,70]. Mutations in myeloid transcription factors (*CEBPA*, *RUNX1*, *GATA2*) also correlated with lack of response to IDHi monotherapy [68]. Acquired mutations in either active signaling or myeloid transcription factor genes were frequently identified at the time of relapse to IVO or ENA [66,68,70] with resultant recapitulation of arrested differentiation of myeloid progenitors [70]. Indeed, leukemic stemness and a hypermethylated phenotype (devoid of mutations in *DNMT3A*) correlated with IDHi resistance [68]. Whether combining IDHi with other targeted or cytotoxic therapies can overcome these described resistance mechanisms observed with IDHi monotherapy and improve outcomes is an area of active investigation [59,71]. Preliminary data suggests upregulation of alternative anti-apoptotic proteins other than BCL-2 (i.e., MCL-1, BCL-xL) may promote resistance in patients treated with the combination of VEN and IVO [71].

## 7. Emerging Therapeutic Targets

Additional agents are in development with promising results in early phase clinical trials. Menin inhibitors disrupt the interaction between menin and *KMT2A* fusion proteins resulting in downregulation of key genes involved in leukemogenesis, including *HOXA* and *MEIS1* [72]. Menin inhibition appears to be most clinically active in patients with *KMT2A*-rearranged (formerly classified as *MLL*-rearranged) or *NPM1*-mutated AML [72,73,74,75]. In a phase 1 dose-escalation study, treatment with revumenib (SNDX-5613) within a heavily pre-treated (median of 4 prior therapies) population of 68 patients with predominantly *KMT2A*-rearranged or *NPM1*-mutated leukemia resulted in a composite CR (CR/CRh/CRp) rate of 38% (N = 23), with slightly higher rates of CR/CRh observed in patients with *KMT2A*-rearranged (33%, N = 15/46) versus *NPM1*-mutated AML (21%, N = 3/14). Twelve patients were successfully bridged to HCT in remission. Notably, the only grade 3 dose-limiting toxicity occurring in the study was QTc prolongation, occurring in 13% of patients; other grade 3 or greater AEs were uncommon [74].

The menin-KMT2A inhibitor ziftomenib (KO-539) also demonstrated safety and efficacy in twelve patients with *KMT2A*-rearranged or *NPM1*-mutated AML [75]. At the escalated 600 mg dose, the CR/CRh rate was 33% (N = 4/12), with 75% of patients attaining MRD negative remissions [75]. Notable AEs ≥ grade 3 occurring in ≥10% of the study population included febrile neutropenia (25%), differentiation syndrome (17%), and diarrhea (17%) [75].

Multiple immunotherapeutic agents targeting the CD47/signal regulatory alpha (SIRPα) axis have been developed for the treatment of AML [76]. In particular, the anti-CD47 antibody magrolimab appears safe with demonstrable activity based on recently presented data from a phase 1b study in patients with AML [77]. In 72 patients with *TP53*-mutated AML that were ineligible for intensive chemotherapy, magrolimab was administered at a final dose of 30 mg/m^2^ weekly or every other week with AZA on D1–7 every 28 days [77]. The CR/CRi rate was 41.6%, with a median time to CR/CRi of 2.2 months [77]. After a median follow-up of 8.3 months, the median OS was 10.8 months [77]. Treatment emergent AEs ≥ grade 3 included febrile neutropenia (37.5%), anemia (29.2%), thrombocytopenia (29.2%), pneumonia (26.4%), and neutropenia (20.8%) [77].

Magrolimab combined with AZA and VEN is also under investigation (NCT04435691) [78]. When used as a frontline combination, AZA+VEN+magrolimab was associated with a CR/CRi rate of 90% and 63% in *TP53* wild-type (N = 10) and mutated (N = 22) patients, respectively [78]. 12-month OS in patients with *TP53* wild-type and mutated AML was estimated at 83% and 53%, respectively [78]. In a difficult-to-treat patient population, anti-CD47 therapy has demonstrated a promising early efficacy signal that warrants validation in larger, ongoing randomized prospective studies (NCT05079230, NCT04778397).

Reactivation of mutant p53 protein with eprenetapopt (APR-246) demonstrated early efficacy in patients with *TP53*-mutated MDS or AML [79,80]. In a phase 1b/2 prospective clinical trial, 44% of participants with MDS or AML attained a CR, with patients harboring *TP53* mutations preferentially attaining CR with eprenetapopt compared to patients with *TP53* wild-type disease (69% vs. 25%); the median OS was 10.8 months [80]. Similar responses were observed in a phase II study conducted by the Groupe Francophone de Myélodysplasies (GFM), where observed CR rates were 47% and 17% in patients with *TP53*-mutated MDS and AML, respectively [79]. The median OS was 12.1 months in the intention-to-treat population [79]. A prospective phase 3 trial of eprenetapopt+AZA vs. AZA was initiated in patients with MDS based on the results of these early studies, but did not meet its primary endpoint, with a CR rate of 33% vs. 22.4% (*p*-value:0.13) [81]. A similar efficacy signal has been observed in the post-HCT setting for eprenetapopt+AZA, which, based on the experience of eprenetapopt in the frontline setting, will require confirmation in larger randomized cohorts [82].

## 8. Conclusions

Molecularly targeted therapy is now a reality for patients with AML. Treatment with FLT3 and IDH1/2 inhibitors serve as proof of principle, with patients deriving benefit from these targeted therapies that improve response rates and survival. Contemporaneous correlative analyses continue to unravel complex mechanisms driving resistance or relapse to these targeted therapies in current practice. These advances will assuredly result in the discovery of new molecular and cellular targets, drive the development of new drug combinations in addition to ongoing early phase trials of emerging therapies, and enable broader use of molecularly-defined treatments to other patient subgroups, thereby improving outcomes in patients with AML.

## Figures and Tables

**Figure 1 cancers-15-01617-f001:**
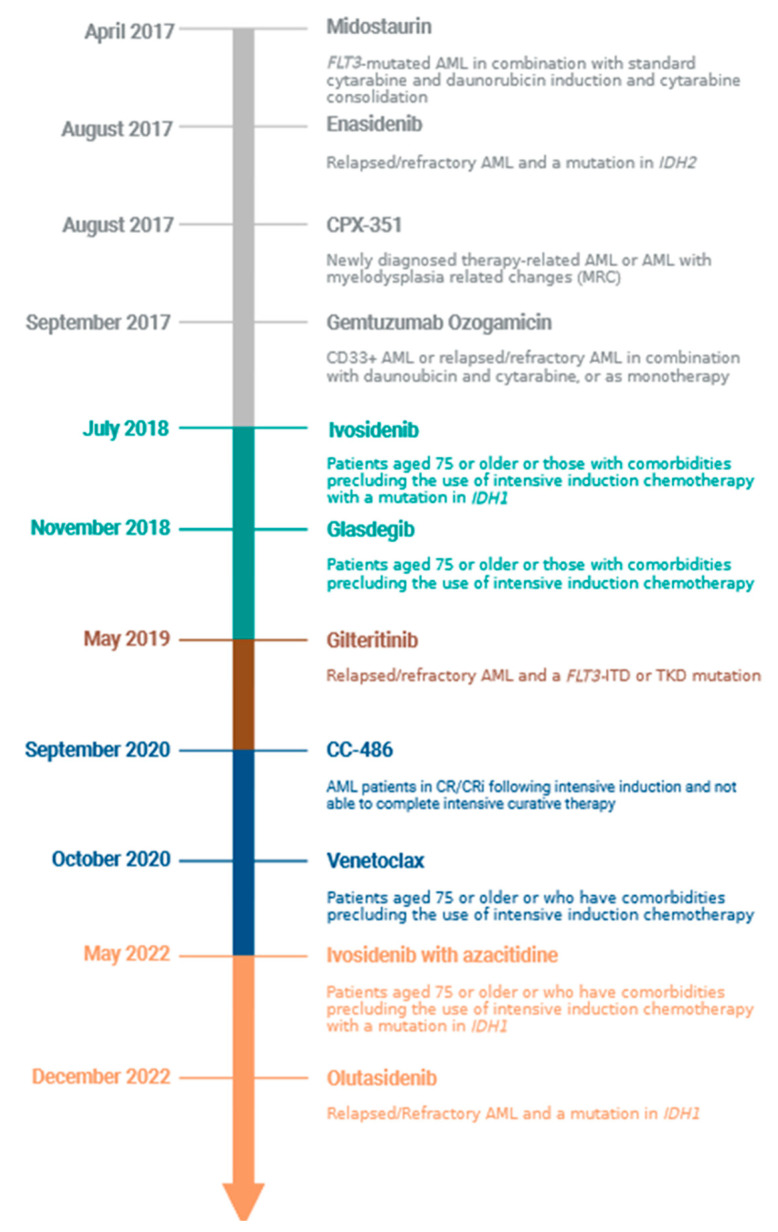
Timeline of FDA-approved agents for AML for specific molecular or clinical subgroups.

**Figure 2 cancers-15-01617-f002:**
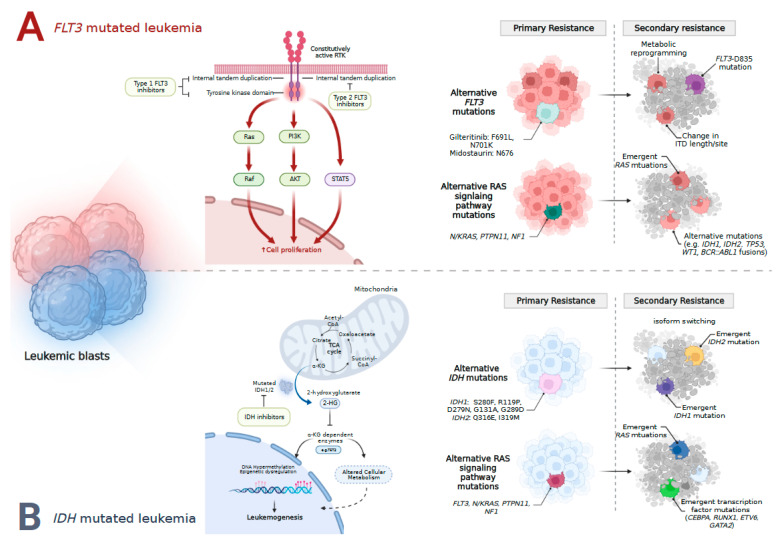
Mechanisms of leukemogenesis in *FLT3*-ITD/TKD-mutated acute myeloid leukemia. Constitutive activation of the FLT3 receptor results in downstream signaling pathway activation promoting cell proliferation and survival. Resistance to FLT3 inhibitor therapy can occur via both primary and secondary resistance mechanisms (**A**). Mechanisms of leukemogenesis in *IDH*-mutated acute myeloid leukemia. Production of the oncometabolite 2-hydroxyglutarate results in epigenetic reprogramming and altered cellular metabolism. As with resistance to FLT3 inhibitors, both primary and secondary resistance mechanisms have been described to IDH inhibitors (**B**).

**Table 1 cancers-15-01617-t001:** Principal studies of FLT3 inhibitors for the treatment of AML.

Study	Patient Population	Chemotherapy Backbone	FLT3 Inhibitor	Response Rate	Survival	Notes
Intensive induction plus sorafenib (SORAML; placebo controlled, randomized phase 2)	ND-AML age ≤ 60 years	Daunorubicin+cytarabine induction and consolidation	Sorafenib (400 mg BID) administered D10–19 (induction) or D2–28 (consolidation, followed by up to 12 months of maintenance therapy	Sorafenib vs. placebo: 60% vs. 59%	3-year EFS: 40% vs. 22%	Fever, diarrhea, bleeding, cardiac events, hand-foot syndrome, rash more common in sorafenib arm
Sorafenib maintenance post alloHCT (SORMAIN; placebo controlled, randomized phase 2)	*FLT3*-ITD-mutated AML in CR following alloHCT	-	Sorafenib 400 mg BID continuous for 24 months post-HCT	-	24-month RFS: 85% vs. 53%; 24-month OS: 90.5% vs. 66.2%	GVHD frequent in both arms (76.8% in sorafenib arm, 59.8% in placebo)
Intensive induction plus midostaurin (RATIFY; placebo controlled, randomized phase 3)	ND-AML age < 60 with *FLT3*-ITD or TKD mutation	Daunorubicin+cytarabine induction and consolidation	Midostaurin 50 mg PO BID D8–21 of induction and consolidation, then continuously for up to 12 months of maintenance therapy	CR:Midostaurin vs. placebo: 58.9% vs. 53.5%	4-year OS: 51.4% vs. 44.3%	Increased risk of rash in midostaurin arm compared to placebo
Intensive induction plus quizartinib (QUANTUM-FIRST; placebo controlled, randomized phase 3)	ND-AML age 18-75 with *FLT3*-ITD mutation	Daunorubicin+cytarabine induction and consolidation	Quizartinib 40 mg PO D8-21 of induction and consolidation, then continuously for up to 36 months of maintenance therapy	CR:Quizartinib vs. placebo: 54.9% vs. 55.4%	Median OS: 31.9 vs. 15.1 months	Similar rates of febrile neutropenia in both arms. QTcF prolongation more common with quizartinib, but QTcF > 500 msec in only 2.3%
Gilteritinib and azacitidine vs. azacitidine (LACEWING; open label, randomized phase 3)	ND-AML with *FLT3*-ITD or TKD mutation ineligible for intensive chemotherapy	Azacitidine 75 mg/m^2^ D1–7 every 28 days	Gilteritinib (120 mg PO daily)	CR/CRh: Gilteritinib+AZA vs. AZA: 25.7% vs. 16.3%	Median OS: 9.82 vs. 8.87 months Median EFS: 0.03 months (both arms)	Febrile neutropenia, pneumonia, and GI hemorrhage more common in GIL+AZA arm. 26% of AEs resulted in death in GIL+AZA arm
Gilteritinib vs. salvage chemotherapy (ADMIRAL; open label, randomized phase 3)	R/R-AML with *FLT3*-ITD or TKD mutation	-	Gilteritinib (120 mg PO daily)	CR/CRh: Gilt vs. salvage chemotherapy: 34% vs. 15.3%	Median OS: 9.3 vs. 5.6 months	Serious AE’s attributed to gilteritinib included febrile neutropenia and elevated AST/ALT
Venetoclax+Gilteritinib (Open label, phase 1b)	R/R AML with FLT3-ITD mutation	Venetoclax 400 mg daily	Gilteritinib (120 mg PO daily)	CRc rate: 75%	Median OS: 10.6 months (no prior FLT3i); 9.6 months (prior FLT3i)	97% of patients experienced Grade 3-4 AE. 80% of patients experienced grade 3-4 cytopenias

ND: newly diagnosed; R/R: relapsed/refractory; PO: per os; BID: twice daily; CR: complete response; CRh: complete response with partial hematologic recovery; CRc: composite complete response (note: CRc may vary by study); EFS: event-free survival; RFS: relapse-free survival; OS: overall survival; GVHD: graft-vs-host-disease; AE: adverse event(s).

**Table 2 cancers-15-01617-t002:** Principal studies of IDH inhibitors for the treatment of AML.

Study	Patient Population	Chemotherapy Backbone	IDH Inhibitor	Response Rate	Survival	Notes
Ivosidenib (open label, phase 1)	R/R-AML with *IDH1* mutation	-	Ivosidenib 500 mg PO daily	CR/CRh: 30%	Median OS: 8.8 months	Grade 3 or greater AE’s included QTc prolongation, IDH differentiation syndrome (IDH-DS), and anemia
Olutasidenib (open label, ongoing phase 2 portion)	R/R-AML with *IDH1* mutation	-	Olutasidenib 150 mg PO BID	CR/CRh: 35%	Median OS: 11.6 months	Grade 3 or greater AE’s included IDH-DS, increased LFTs, and QTc prolongation
Ivosidenib+azacitidine vs. azacitidine (AGILE; placebo controlled, randomized phase 3)	ND-AML with *IDH1* mutation	Azacitidine 75 mg/m^2^ D1–7 every 28 days	Ivosidenib 500 mg PO daily	CR/CRi: IVO+AZA vs. AZA: 51% vs. 18%	Median OS: 24 vs. 7.9 months	IVO+AZA had less infectious complications compared to AZA (28% vs. 49%) but increased hemorrhagic events (41% vs. 29%). IDH-DS occurred in 14%
Enasidenib (open label, phase 1/2)	R/R-AML with *IDH2* mutation	-	Enasidenib 100 mg daily	CR: 19.3%	Median EFS: 6.4 months Median OS: 9.3 months	Common treatment-related AEs included hyperbilirubinemia (8%), IDH-DS (11%), anemia (10%), thrombocytopenia (8%), and TLS (5%)
Enasidenib vs. conventional care regimens (IHDENTIFY; open label, randomized phase 3)	R/R-AML with *IDH2* mutation	-	Enasidenib 100 mg daily	CRc: ENA vs. CCR 29.7% vs. 6.2%	Median OS: 6.5 vs. 6.2 months	
Enasidenib+azacitidine vs. azacitidine (open label, phase 1b)	ND-AML with *IDH2* mutation	Azacitidine 75 mg/m^2^ D1–7 every 28 days	Enasidenib 100 mg daily	CR/CRi: ENA+AZA vs. AZA (63% vs. 30%)	Median EFS: 15.9 vs. 11.9 months Median OS: 22 vs. 22.3 months	Grade 3–4 neutropenia (37% vs. 25%) and thrombocytopenia (37% vs. 19%) numerically higher with ENA+AZA
Enasidenib or ivosidenib with 7+3 induction and consolidation (open label, phase 1)	ND-AML with either *IDH1* or *IDH2* mutation	Daunorubicin and cytarabine induction and consolidation (cytarabine or mitoxantrone/etoposide)	Ivosidenib 500 mg PO daily Enasidenib 100 mg PO daily	Composite CR: 77% (IVO), 74% (ENA)	Median OS: not reached (IVO; 12-month OS 78%), 25.6 months ENA	Toxicity profile similar to that observed with 7+3 induction and consolidation

ND: newly diagnosed; R/R: relapsed/refractory; PO: per os; BID: twice daily; CR: complete response; CRh: complete response with partial hematologic recovery; CRc: composite complete response (note: CRc may vary by study); EFS: event-free survival; OS: overall survival; AE: adverse event(s); IDH-DS: IDH differentiation syndrome.

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
