# Peer review of "Molecularly Targeted Therapy in Acute Myeloid Leukemia: Current Treatment Landscape and Mechanisms of Response and Resistance"

_cancers, 2023, doi:10.3390/cancers15051617_

Round 1
Reviewer 1 Report
This review article by Curtis et al is a comprehensive, well-written, and up-to-date analysis that adequately describes the field of molecular targeted therapy in AML.
I think the authors did a great job summarizing a great amount of information.
My only suggestion would be that at the beginning of the article, the authors could indicate, by writing a brief paragraph, that molecular targeted therapy is one of different alternatives for AML treatment. Other alternatives are hematopoietic transplants and immunotherapy.
This is just a suggestion. It is up to the authors to include it.
Author Response
This review article by Curtis et al is a comprehensive, well-written, and up-to-date analysis that adequately describes the field of molecular targeted therapy in AML.I think the authors did a great job summarizing a great amount of information.
- My only suggestion would be that at the beginning of the article, the authors could indicate, by writing a brief paragraph, that molecular targeted therapy is one of different alternatives for AML treatment. Other alternatives are hematopoietic transplants and immunotherapy. This is just a suggestion. It is up to the authors to include it.
Thank you for this comment. We have rephrased the introduction to highlight that hematopoetic cell transplantation remains the only curative therapy for AML. We also discuss immunotherapy within the context of the anti-CD47 antibodies later in the text to highlight the potential role for immunotherapy in AML.
The following has been added to the introductory paragraph:
“While consolidative hematopoietic cell transplantation in remission remains the only curative therapy for AML, these new treatment options are a welcome addition to a therapeutic repertoire that has largely relied upon standard cytotoxic chemotherapy for the last 40 years3,5-9.”
Reviewer 2 Report
Review: Molecularly targeted therapy in Acute Myeloid Leukemia: Current treatment landscape and mechanisms of response and resistance
Lachowiez et al. provide an overview of the targeted therapies in AML. This review is organized in various chapters describing the different mutational landscape in AML and their specific directed therapies, with the focus on FLT3 and IDH. It also describes potential resistance mechanisms and an outlook on future therapeutic targets.
Overall, this review is well-written and comprehensive.
Minor comments:
- all abbreviations should be explained at least once in the text. Since this review summarized a lot of clinical trials, there are many abbreviations appearing, and not all of them are explained. Missing explanations for abbreviations include CR, HR, CI, BID, PO, ND, QT, CRi, CRh, NR, ANC, ALT, AST, VAF, LFT, MRD, PCR, MFC, NR
- apart from the main text, the abbreviations should also be explained again in the table and figure legends, to make them understandable without having to read through the whole manuscript.
- the numbering of the different paragraphs is very confusing and not well organized. I would suggest doing the following or something similar:
1. Introduction
2. FLT3 directed therapy in ND-AML (including the general paragraph of FLT3 mutations)
2.1. Sorafenib
…
2.7. Triplet regimens in ND-AML
3. FLT3 mutations in R/R-AML
3.1. Gilteritinib
…
3.4. FLT3 mechanisms of resistance
4. IDH mutations
4.1. IVO
…
4.8. Mechanisms of resistance
5. Emerging targets
6. Conclusions
- Figure 1: there are agents named that don’t appear in the text (e.g.CPX-351, CC-486). They should either be explained in the Figure or in the main text. Also, it would be helpful to at least briefly mention in the figure what the different agents target.
- Table 1: second last row, 5th column: the percentage is missing
- Figure 2: the figure legend should be revised.
- there are several typos:
Line 40, page 2: Parallel, an increased…
Line 50, page 3: … mutations are associated with…
Line 106 and 107, page 4: OS compared
Line 115, page 4: OS rate
Line 161, page 5: GIL + AZA
Line 331, page 9: Table 2:…of AML.
Line 422, page 11: (CCR;…).
Line 430, page 11: [95% CI…].
Author Response
Lachowiez et al. provide an overview of the targeted therapies in AML. This review is organized in various chapters describing the different mutational landscape in AML and their specific directed therapies, with the focus on FLT3 and IDH. It also describes potential resistance mechanisms and an outlook on future therapeutic targets. Overall, this review is well-written and comprehensive.
- Minor comments:
- All abbreviations should be explained at least once in the text. Since this review summarized a lot of clinical trials, there are many abbreviations appearing, and not all of them are explained. Missing explanations for abbreviations include CR, HR, CI, BID, PO, ND, QT, CRi, CRh, NR, ANC, ALT, AST, VAF, LFT, MRD, PCR, MFC, NR. Apart from the main text, the abbreviations should also be explained again in the table and figure legends, to make them understandable without having to read through the whole manuscript.
Thank you. These abbreviations have been defined within the manuscript text in addition to the legend text for the tables/figures as well.
- The numbering of the different paragraphs is very confusing and not well organized. I would suggest doing the following or something similar:
- Introduction
- FLT3 directed therapy in ND-AML (including the general paragraph of FLT3 mutations)
2.1. Sorafenib
…
2.7. Triplet regimens in ND-AML
- FLT3 mutations in R/R-AML
3.1. Gilteritinib
…
3.4. FLT3 mechanisms of resistance
- IDH mutations
4.1. IVO
…
4.8. Mechanisms of resistance
- Emerging targets
- Conclusions
Thank you for this suggestion. We have revised the subsections of text to be more conducive to the reader as also suggested by reviewer #4.
- Figure 1: There are agents named that don’t appear in the text (e.g.CPX-351, CC-486). They should either be explained in the Figure or in the main text. Also, it would be helpful to at least briefly mention in the figure what the different agents target.
Thank you. We have updated the text to include an additional description of CPX-351 and CC-486 as they relate to the manuscript. We agree that what each agent targets is important, however the description listed in figure 1 is the FDA labeled indication for each agent which describes the population for which it is approved.
The following has been added to the manuscript:
“AML treatment historically relied on anthracycline and cytarabine based regimens3,4, however since 2017 the treatment landscape has expanded to include eleven additional FDA approved medications and/or regimens, including eight targeting specific molecular or cellular subgroups Figure 1. Though not specific for a molecular or cellular target, CPX-351 demonstrated efficacy in patients with secondary AML or AML with myelodysplasia related changes and oral azaciditine (CC-486) improved survival in patients with NPM1 and/or FLT3-mutated AML, further supporting the ability to individualize therapy based on clinical characteristics in addition to cytogenetic or molecular features84-86.
- Table 1: second last row, 5th column: the percentage is missing
- Figure 2: the figure legend should be revised.
- there are several typos:
Line 40, page 2: Parallel, an increased…
Line 50, page 3: … mutations are associated with…
Line 106 and 107, page 4: OS compared
Line 115, page 4: OS rate
Line 161, page 5: GIL + AZA
Line 331, page 9: Table 2:…of AML. ***
Line 422, page 11: (CCR;…).
Line 430, page 11: [95% CI…].
Thank you for highlighting these grammatical errors. These have been corrected and the manuscript reviewed for additional errors.
Reviewer 3 Report
The manuscript "Molecularly targeted therapy in Acute Myeloid Leukemia: Current treatment landscape and mechanisms of response and resistance" is an interesting review about the principal clinical trials and emerging therapies in AML. The work is well written and scientifically sound, collecting a broad range of data from studies of FLT3 and IDH inhibitors. Thanks to the in-depth discussion of doses, CR and survival related to different types of patient population and mutations, the work is suitable for publication after the following minor revisions:
- Merge paragraph 1.2 with 1.1 and add more informations about the different classifications. In the current state, paragraph 1.2 is sparse.
- Some clinical trials described in the text, are not reported in table 1 or 2. Add them to improve data presentation.
- Table 1 has two different captions (above and below). Remove one of them.
- In table 1, the response rate values for the ADMIRAL study are missing.
- References 2 and 3 are incomplete. Furthermore, lines 34 and 36 have the same citations, some of which are too old. Change them with examples of recent advances and studies for the treatment of AML, proving the efforts in the field: Drug Dev Res. 2022 Sep;83(6):1331-1341. doi: 10.1002/ddr.21962. Epub 2022 Jun 24. PMID: 35749723; J. Med. Chem. 2020, 63, 21, 12403–12428 https://doi.org/10.1021/acs.jmedchem.0c00696
Author Response
The manuscript "Molecularly targeted therapy in Acute Myeloid Leukemia: Current treatment landscape and mechanisms of response and resistance" is an interesting review about the principal clinical trials and emerging therapies in AML. The work is well written and scientifically sound, collecting a broad range of data from studies of FLT3 and IDH inhibitors. Thanks to the in-depth discussion of doses, CR and survival related to different types of patient population and mutations, the work is suitable for publication after the following minor revisions:
- Merge paragraph 1.2 with 1.1 and add more informations about the different classifications. In the current state, paragraph 1.2 is sparse.
Thank you. We have reformatted the introduction to convey the necessary aspects pertinent to the readership prior to the beginning of the manuscript and expanded upon the most recent updates to the ELN classification system.
The following has been added to the text:
“Acute myeloid leukemia (AML) is an aggressive hematologic malignancy characterized by arrested differentiation of hematopoietic stem and progenitor cells resulting from alterations in a heterogeneous array of genomic drivers1. Characterization of this genomic diversity led to the recognition and refinement of prognostic and predictive molecular markers, enabling risk stratification in AML1,2. The most recent iteration of the European LeukemiaNet (ELN) guidelines recognizes not only the importance of genomic characterization in assignment of disease risk, but the additional role of molecularly targeted therapy (namely IDH and FLT3 inhibitors) in the treatment of patients with AML2.
- Some clinical trials described in the text, are not reported in table 1 or 2. Add them to improve data presentation.
Thank you for this suggestion. While there are several retrospective analyses mentioned in the manuscript, we have clarified in the text that the tables provided include only prospective trials in patients receiving FLT3 or IDH inhibitors, in order to provide the most pertinent information to the reader.
- Table 1 has two different captions (above and below). Remove one of them.
Thank you. This has been updated.
- In table 1, the response rate values for the ADMIRAL study are missing.
Thank you. This has been updated.
- References 2 and 3 are incomplete. Furthermore, lines 34 and 36 have the same citations, some of which are too old. Change them with examples of recent advances and studies for the treatment of AML, proving the efforts in the field: Drug Dev Res. 2022 Sep;83(6):1331-1341. doi: 10.1002/ddr.21962. Epub 2022 Jun 24. PMID: 35749723; J. Med. Chem. 2020, 63, 21, 12403–12428 https://doi.org/10.1021/acs.jmedchem.0c00696
Thank you. We have added the complete citations for references 2 and 3. We appreciate the reviewer providing additional references to update the text. However, the citations included in line 34-36 were chosen purposefully as they represent the seminal studies in the field of AML combining cytotoxic chemotherapy, highlighting the lack of progress until more recent advances.
Reviewer 4 Report
This review article focuses on molecularly targeted therapies in AML. It is quite thorough with a lot of data included. The article is really focused primarily on FLT3 and IDH mutations, and would be worth clarifying this in the abstract and introduction.
The organization of the article is somewhat confusing and inconsistent. For FLT3 section, would reorganize to discuss each FLT3 inhibitor first (in both newly diagnosed and r/r setting) and then combinations. As it stands now, the authors discuss combinations with gilterinitib prior to discussing the drug itself, which is confusing. Section 2 is headed as “relapsed/refractory AML”, however, discussion of IDH inhibitors, including for newly diagnosed patients, is under this section. Would suggest to make section 1 about FLT3 (newly diagnosed, then r/r, then resistance mechanisms), section 2 re: IDH mutations (newly diagnosed, then r/r, then resistance mechanisms) and section 3 on emerging targets, or another similar layout that is consistent and follows for the entire article. Similarly, the subheadings should be consistent. For example “ivosidenib with azacitidine” should be under “ivosidenib in the treatment of IDH1 mutated AML” and not its own category. Would consider a brief section on integration of targeted therapies for pediatric AML.
There are a few discrepancies with numbers of drugs. Please check these throughout the manuscript. For example, authors state that there are 10 new approvals since 2017 but Figure 1 shows 11. In Line 329- authors state that 2 IDH inhibitors are approved. Should be 3? (there are three listed in Figure 1)
For figure 1 timeline, I would suggest adding if the drug is approved in combination with other agents (for example for venetoclax and glasdegib). When were the prior approvals for drugs for AML? May be interesting to highlight the gap between prior approvals and the more recent “high” number of approvals for drugs for aml.
For midostaurin efficacy, the authors report that midostaurin was better for all FLT3 subgroups. However this is not supported by the Stone et al NEJM report (Figure 2), where there was a trend but not statistically significant difference for midostaurin benefit. Would clarify this in the manuscript.
Minor points/typos:
- Line 40: “Parallel” is out of place
- Line 63: FLT3i is extra (after parentheses)
- Line 504—lead sentence starts with “mechanisms of single agent IVO have been described…” but then discusses both IDH1 and IDH2 mutations, and paragraph leads more with IDH2. Should clarify first sentence.
- Line 553: should be “SNDX”
Author Response
- This review article focuses on molecularly targeted therapies in AML. It is quite thorough with a lot of data included. The article is really focused primarily on FLT3 and IDH mutations, and would be worth clarifying this in the abstract and introduction.
Thank you. We have updated the abstract and introduction to state emphasize that the review is predominantly focused on FLT3 and IDH inhibitors. The following changes have been made:
Abstract:
“Herein, we comprehensively review the current landscape of IDH and FLT3 inhibitors in clinical practice for the treatment of AML, discuss resistance mechanisms and identified cellular/molecular targets for development of new combination therapies, and cellular and molecularly targeted therapies currently in ongoing early phase clinical trials.”
Introduction:
“Herein, we review the contemporary landscape of molecularly targeted therapies in AML with a focus on FLT3 and IDH inhibitors, identified mechanisms of resistance to these agents, and early investigations of therapies utilizing molecularly targeted agents in combination or with cytotoxic chemotherapy.”
- The organization of the article is somewhat confusing and inconsistent. For FLT3 section, would reorganize to discuss each FLT3 inhibitor first (in both newly diagnosed and r/r setting) and then combinations. As it stands now, the authors discuss combinations with gilterinitib prior to discussing the drug itself, which is confusing. Section 2 is headed as “relapsed/refractory AML”, however, discussion of IDH inhibitors, including for newly diagnosed patients, is under this section. Would suggest to make section 1 about FLT3 (newly diagnosed, then r/r, then resistance mechanisms), section 2 re: IDH mutations (newly diagnosed, then r/r, then resistance mechanisms) and section 3 on emerging targets, or another similar layout that is consistent and follows for the entire article. Similarly, the subheadings should be consistent. For example “ivosidenib with azacitidine” should be under “ivosidenib in the treatment of IDH1 mutated AML” and not its own category. Would consider a brief section on integration of targeted therapies for pediatric AML.
Thank you. We have reformatted the article based on these recommendations to clarify the sections to the reader. As the article is focused on the treatment of adult patients with AML, pediatric literature was not included, but is an interesting idea for another review covering this topic.
- There are a few discrepancies with numbers of drugs. Please check these throughout the manuscript. For example, authors state that there are 10 new approvals since 2017 but Figure 1 shows 11. In Line 329- authors state that 2 IDH inhibitors are approved. Should be 3? (there are three listed in Figure 1)
Thank you. This has been corrected.
- For figure 1 timeline, I would suggest adding if the drug is approved in combination with other agents (for example for venetoclax and glasdegib). When were the prior approvals for drugs for AML? May be interesting to highlight the gap between prior approvals and the more recent “high” number of approvals for drugs for aml.
Thank you for this comment. We have updated the figure to include the drug partner for those drugs that were approved in combination as recommended. We appreciate the comment of inclusion of past approvals and the extended period prior to more recent drug developments in AML, however as the review is focused on new approvals, we have elected to focus the figure only on the more recent drug approvals for AML.
- For midostaurin efficacy, the authors report that midostaurin was better for all FLT3 subgroups. However this is not supported by the Stone et al NEJM report (Figure 2), where there was a trend but not statistically significant difference for midostaurin benefit. Would clarify this in the manuscript.
Thank you for this suggestion. We agree as no subgroup demonstrated a significant benefit, but rather FLT3-ITD/TKD mutations taken as a group. We have updated the manuscript accordingly. The following has been added to the text:
“The benefit of midostaurin was observed across all FLT3-mutated patients (i.e., including patients with FLT3-ITD and/or TKD mutations; a trend in favor of midostaurin was observed in subgroup analyses of individual FLT3 variants) consistent with its known activity against either FLT3 variant.
- Minor points/typos:
- Line 40: “Parallel” is out of place
- Line 63: FLT3i is extra (after parentheses)
- Line 504—lead sentence starts with “mechanisms of single agent IVO have been described…” but then discusses both IDH1 and IDH2 mutations, and paragraph leads more with IDH2. Should clarify first sentence.
- Line 553: should be “SNDX
Thank you. These have been corrected.